# Fatty acid metabolism suppresses neonatal cardiomyocyte proliferation by increasing PDK4 and HMGCS2 expression through PPARδ

Shota Tanaka[1]*, Akane Hirota[1], Yoshiaki Okada[1], Masanori Obana[1,2], Yasushi Fujio[1,2]

1 Laboratory of Clinical Science and Biomedicine, Graduate School of Pharmaceutical Sciences, Osaka University, Osaka, Japan, 2 Integrated Frontier Research for Medical Science Division, Institute for Open and Transdisciplinary Research Initiatives, Osaka University, Osaka, Japan

* tanaka-s@phs.osaka-u.ac.jp

## Abstract

Cardiomyocytes lose their capacity to regenerate immediately after birth. Simultaneously, cardiomyocytes change energy metabolism from glycolysis to oxidative phosphorylation, especially using fatty acids. Accumulating evidence has revealed that fatty acid metabolism weakens the proliferative ability of cardiomyocytes. However, its underlying molecular mechanism remains unclear. In this study, we investigated how fatty acid metabolism contributes to cell cycle regulation in neonatal cardiomyocytes. Cultured neonatal rat cardiomyocytes (NRCMs) were treated with a fatty acid mixture (FA) consisting of palmitic and oleic acids containing L-carnitine. The FA treatment increased not only β-oxidation-related enzymes but also pyruvate dehydrogenase kinase 4 (PDK4), a fatty acid metabolism regulator, and HMG-CoA synthase 2 (HMGCS2), a ketogenic factor. Moreover, Ki67-positive proliferative NRCMs were reduced by the FA, indicating that fatty acids suppress the NRCM cell cycle. GW501516, a peroxisome proliferator-activated receptor δ (PPARδ) activator, also upregulated fatty acid metabolism genes and disturbed NRCM proliferation, whereas GSK3787, a PPARδ inhibitor, recovered FA-induced the cell cycle arrest. Furthermore, overexpression of PDK4 or HMGCS2 using a lentiviral vector suppressed cell cycle activity in NRCMs, and silencing either gene regained cell cycle even in FA-rich condition. In conclusion, fatty acid metabolism increased PDK4 and HMGCS2 via PPARδ activation and suppressed NRCM proliferation.

## Introduction

Heart failure is the primary cause of mortality worldwide. Heart failure is the end stage of many cardiac diseases, such as cardiomyopathy, angina, and myocardial infarction. Almost all cases of heart failure get worse because cardiomyocytes rarely regenerate to recover damaged hearts. Therefore, promoting cardiomyocyte regeneration may be a breakthrough method for treating cardiac diseases.

**Data availability statement:** All relevant data are within the manuscript and its Supporting Information files.

**Funding:** This study is partially supported by MEXT/JSPS KAKENHI Grants 20K22707 to ST, 22K15277 to ST. This research was also supported by Basis for Supporting Innovative Drug Discovery and Life Science Research (BINDS) from AMED under grant numbers JP23ama121052 and JP23ama121054.

**Competing interests:** The authors have no conflicts of interest associated with this manuscript.

In many species of fish and amphibians, such as zebrafish and western clawed frogs, cardiomyocytes retain their regenerative activity throughout their lifetime [1, 2]. In contrast, cardiomyocytes of mammals, including mice, pigs, and humans, have proliferative ability immediately after birth but exit the cell cycle [3–8]. A previous study showed that mice lose the capacity for cardiac regeneration by seven days after birth, based on the finding that a 1-day-old mouse could completely repair the heart whose apex was excised; however, a 7-day-old mouse could not [4]. Cardiac metabolic changes are also associated with cardiac growth in mammals. In particular, fetal and perinatal hearts generate energy by anaerobic respiration; however, adult hearts perform aerobic respiration using fatty acids, mainly because fatty acid utilization produces more energy than anaerobic glycolysis [9–11].

Long-chain fatty acids are taken into cells through passive diffusion or fatty acid translocase (FAT/CD36), followed by transportation into the mitochondria by carnitine palmitoyltransferase (CPT) 1 and 2. In the mitochondrial matrix, fatty acids are digested into acetyl-CoA through β-oxidation using long-chain acyl-CoA dehydrogenase (LCAD), medium-chain acyl-CoA dehydrogenase (MCAD), and so forth. Acetyl-CoA is primarily used in the tricarboxylic acid cycle to synthesize ATP. The cardiac expression of enzymes involved in fatty acid metabolism is gradually upregulated after birth, whereas that of glycolysis-related enzymes is downregulated [10]. Therefore, it is likely that the metabolic change from glycolysis to fatty acid utilization is closely related to the proliferative activity of cardiomyocytes. However, it remains to be fully elucidated how fatty acid metabolism regulates cardiomyocyte proliferation because there are various lines of experimental evidence. For example, neonatal mice fed fatty acid-deficient milk showed a delay in cardiomyocyte cell cycle arrest [12]. Moreover, pyruvate dehydrogenase kinase 4 (PDK4) knockout mice which have reduced fatty acid utilization showed increased cardiomyocyte proliferation after myocardial infarction [12]. On the other hand, the activation of β-oxidation temporally promoted cardiomyocyte proliferation immediately after birth [13].

Ketone bodies are made from acetyl-CoA excessively produced by β-oxidation in hepatic mitochondria and transported to the heart as an energy resource [14]. β-hydroxybutyrate, a representative ketone body, is synthesized from acetyl-CoA through the HMG-CoA synthase 2 (HMGCS2)/HMG-CoA lyase (HMGCL)/β-hydroxybutyrate dehydrogenase 1 (BDH1) axis. Chong et al. reported that ketone body promoted the maturation of cardiomyocytes and cell cycle arrest [10] but Cheng et al. showed that ketogenesis induced cardiac regeneration in injured heart [15]. Thus, the relationship between ketogenesis and cardiomyocyte proliferation is controversial.

In this study, we investigated how fatty acid metabolism regulated the cardiomyocyte cell cycle and found that fatty acid metabolism suppresses cardiomyocyte proliferation by increasing PDK4 and HMG-CoA synthase 2 (HMGCS2) expression via activating peroxisome proliferator activator receptor δ (PPARδ). This study demonstrates that increased PDK4 and HMGCS2 expressions are essential for fatty acid-induced cardiomyocyte cell cycle arrest.

## Materials and methods

### Preparation of cardiomyocytes from neonatal rats

Animal experiments were approved by the Experimental Animal Care and Use Committee of the Graduate School of Pharmaceutical Sciences, Osaka University (approved as Douyaku R01-1–6). Animal experiments were performed according to the Guide for the Care and Use of Laboratory Animals, Eighth Edition, updated by the US National Research Council Committee in 2011. Cardiomyocytes were prepared according to the protocol described previously [16]. Briefly, 0-day-old Wistar rats (Kiwa Laboratory Animals) were sacrificed by harvesting their hearts under anesthesia using isoflurane (Wako) without sexual distinction. The scarification was performed in accordance with American Veterinary Medical Association Guidelines for the Euthanasia of Animals: 2020 Edition. The hearts were digested with a mixture of 0.1% collagenase type IV (Sigma-Aldrich) and 0.1% trypsin (Thermo Fisher Scientific). Isolated cells were centrifuged at 300 × g for 5 min, resuspended in DMEM containing 10% FBS (Thermo Fisher Scientific) and 1% penicillin/streptomycin (Nacalai Tesque), and seeded in 10 cm cell culture dishes (Iwaki). To remove non-cardiomyocytes, the medium was collected and centrifuged at 200 × g for 5 min 2 h after seeding. Cardiomyocytes were seeded on a 0.1% gelatine-coated plate and cultured in the incubator (37 °C, 5% $CO_2$/ 95% air). The medium was replaced with 1% FBS 24 h after seeding.

### Fatty acids treatment

The cells were treated with a fatty acid mixture (FA) referring to a previous study [17]. FA contained palmitic acid, oleic acid, and L-carnitine (Nacalai Tesque) at a 1:1:2 molar ratio in 1% FBS-DMEM. The concentrations of the FA indicate the total concentrations of palmitic and oleic acids. Namely, the 500 µM FA contained 250 µM palmitic acid, 250 µM oleic acid, and 500 µM L-carnitine.

### Reagents

The reagents used were fenofibrate, pioglitazone (Wako), GW501516, GW6471, T0070907, and GSK3787 (Selleck Biotechnology).

### Real-time RT-PCR

Real-time RT-PCR was performed as described previously [18]. Briefly, total RNA was extracted using QIAzol (Qiagen), and cDNA was synthesized using oligo dT primer (Thermo Fisher Scientific) and ReverTra Ace (Toyobo). The amount of cDNA was quantified using the Applied Biosystems StepOne Real-Time PCR system (Applied Biosystems) and Fast SYBR Green Master Mix (Thermo Fisher Scientific). The mRNA expression was normalized to that of Gapdh. Primer sequences used for real-time RT-PCR are listed in Table 1.

### Western blotting

Western blotting was performed as described previously [18]. Briefly, the proteins were separated by SDS-PAGE and transferred to an Immobilon-P PVDF membrane (Merck Millipore), followed by blocking with 5% BSA (Nacalai Tesque) for 1 h. After being incubated with primary antibodies overnight at 4 °C, the membranes were reacted with proper secondary antibodies conjugated with horseradish peroxidase (HRP) for 2 h at room temperature. After developing the target protein with ECL reagent (Promega), light emission was detected with ImageQuant LAS 4010 using ImageQuant TL software (GE Healthcare). Protein bands were quantified using ImageJ software (National Institutes of Health). The protein expression was normalized to that of GAPDH. The antibodies used in this study were as follows; mouse anti-GAPDH (1:4000, Merck Millipore, Cat# MAB374, RRID: AB_2107445), rabbit anti-PDK4 (1:500, Proteintech, Cat# 12949–1-AP, RRID: AB_2161499), rabbit anti-HMGCS2 (1:500, abcam, Cat# ab137043, RRID:AB_2749817), mouse anti-Luciferase (1:1000, Santa Cruz Biotechnology, Cat# sc-74548, RRID: AB_1125118), and rabbit anti-γH2AX (1:250, Cell Signaling Technology,

**Table 1. Primers used for real-time RT-PCR.**

| target genes | | oligonucleotide sequense (5' → 3') |
|---|---|---|
| Cd36 | forward | AGATGCAGCCTCCTTTCCAC |
| | reverse | TGTCCAGCACACCATACGAC |
| Cpt1b | forward | TTCCTGGACGAGGTGCTTTC |
| | reverse | TTCCTGGACGAGGTGCTTTC |
| Lcad | forward | CCGCCCGATGTTCTCATTCT |
| | reverse | GACAACAAGTCCCCACCGAT |
| Mcad | forward | AGCCCTGGACGAAGCTACTA |
| | reverse | GCGAGCTGGTTGGCAATATC |
| Glut1 | forward | CTTTGAAGTAGGCCCCGGTC |
| | reverse | GCCACACAGTTGCTCCACA |
| Glut4 | forward | TCCAGTATGTTGCGGATGCT |
| | reverse | CCACCATTTTGCCCCTCAGT |
| Hk2 | forward | CTCAGATAGAGAGCGACTGCC |
| | reverse | GCCCACTGTCACTTTGAGGT |
| Pdk4 | forward | CCGTTGACCCCGTTACCAAT |
| | reverse | GCACACTCAAAGGCATCTTCG |
| Hmgcs2 | forward | GAACTTCTCTCCCCCTGGTG |
| | reverse | GAATGGTTGTATGGATTGGCCTC |
| Hmgcl | forward | TCAGAAGTTTCCCGGCATCAA |
| | reverse | ACAGGAGACATACCCTCTCAC |
| Bdh1 | forward | GCAACAGTGAGGAGGTGGAG |
| | reverse | TAACAACACGGCCTTTGGCT |
| Myh6 | forward | CCTCAAACTCATGGCCACAC |
| | reverse | TGTTCAGATTTTCCCGGTGGA |
| Myh7 | forward | AGGAAGAACCTACTGCGACTG |
| | reverse | CTACTCTTCATTCAGGCCCTTG |
| Gapdh | forward | CATCACCATCTTCCAGGAGCG |
| | reverse | GAGGGGCCATCCACAGTCTTC |

Cat# 9718, RRID: AB_10121789) as primary antibodies and HRP-goat anti-mouse IgG (1:4000, Jackson ImmunoResearch, Cat# 115-035-062, RRID: AB_2338504) and HRP-goat anti-rabbit IgG (1:1000, Cell Signaling Technology, Cat# 7074, RRID: AB_2099233) as secondary antibodies.

## Immunofluorescent microscopic analysis

NRCMs were fixed with 4% paraformaldehyde (Nacalai Tesque) for 15 min and permeabilized with 0.4% triton-X100 in PBS for 5 min. After blocking with 1% BSA for 1 h, the NRCMs were incubated with primary and secondary antibodies for 2 h at room temperature. DAPI (5 μg/mL, Nacalai Tesque) was used to counterstain the nucleus simultaneously along with the secondary antibodies. Fluorescent images were captured using a Cell Voyager CV8000 (Yokogawa Electric) or BZ-X710 (Keyence). The proportion of Ki67-positive NRCMs was calculated using a blank test. The antibodies used in this study were as follows; rat anti-Ki67 (1:2000, Thermo Fisher Scientific, Cat# 14-5698-80, RRID: AB_10853185), rabbit anti-phospho-Histone H3 (Ser10) (1:250, Cell Signaling Technology, Cat# 9701, RRID: AB_331535), rabbit anti-Aurora B (1:250, abcam, Cat# ab2254, RRID: AB_302923), rabbit anti-γH2AX (1:250, Cell Signaling Technology, Cat# 9718, RRID: AB_10121789), and mouse anti-α-actinin (1:1000, Sigma-Aldrich, Cat# A7811, RRID: AB_476766) as primary antibodies

and Alexa Fluor 546-goat anti-rat (1:1000, Thermo Fisher Scientific, Cat# A-11081, RRID: AB_2534125), Alexa Fluor 546-goat anti-rabbit (1:1000, Thermo Fisher Scientific, Cat# A-11035, RRID:AB_143051), and Alexa Fluor 488-goat anti-mouse (1:1000, Thermo Fisher Scientific, A28175, RRID: AB_2536161) as secondary antibodies.

## Quantification of acetyl-CoA

Acetyl-CoA levels were analyzed with RH-NH2 (Wako). NRCMs were reacted with 5 µM RH-NH2 for 30 min at 37 °C and the fluorescent intensity was measured by GloMax Explorer Multimode Microplate Reader (Promega). The excitation and emission wavelengths were 520 nm and 580–640 nm.

## Quantification of ATP

ATP was measured with ATP Assay Kit-Luminescence (Dojindo), according to the manufacturer's protocol. Briefly, NRCMs were reacted with working solution which contained D-Luciferin and Luciferase for 10 min at 37 °C, followed by the measurement of the luminescence using SpectraMax M5e (Molecular Devices).

## Quantification of β-hydroxybutyrate

β-hydroxybutyrate was measured with β-Hydroxybutyrate (Ketone Body) Colorimetric Assay Kit (Cayman), according to the manufacturer's protocol. Briefly, NRCMs were collected and sonicated 10 times, followed by centrifugation at 10000 × g for 10 min. The pellets were resuspended with 110 µL Assay Buffer and 50 µL the resuspended buffer were reacted with Developer Solution for 30 min at 25 °C by duplicate. The absorbances at 450 nm were measured by GloMax Explorer Multimode Microplate Reader.

## Quantification of reactive oxygen species (ROS)

ROS levels were analyzed using CellROX Green (Thermo Fisher Scientific), according to the manufacturer's protocol. Briefly, NRCMs were reacted with 5 µM CellROX Green for 30 min at 37 °C, and the fluorescent intensity was measured by GloMax Explorer Multimode Microplate Reader. The excitation and emission wavelengths were 475 nm and 500–550 nm.

## Cell proliferation assay

NRCM proliferation was measured using Cell Counting Kit-8 (Dojindo), according to the manufacturer's protocol. Briefly, NRCMs were reacted with 10 µL WST-8 containing solution and incubated for 1 h at 37 °C. The absorbances at 450 nm were measured by GloMax Explorer Multimode Microplate Reader.

## Plasmids

Cardiac troponin T (cTNT) promoter-inserted lentivirus packaging plasmid to express Venus was made by replacing the EF-1α promoter of pCSII-EF-Venus [19] with the cTNT promoter. The insert DNA was amplified from pAAV9:cTNT::3Flag-hYAP (S127A) by PCR using Q5 Hot Start High-Fidelity DNA Polymerase (New England Biolabs), according to the manufacturer's protocol and inserted into AgeI-digested pCSII-EF-Venus using Ligation High Ver. 2 (Toyobo). Luciferase, PDK4, and HMGCS2 expressing packaging plasmids were constructed by replacing the Venus site of pCSII-cTNT-Venus with the respective sequences. The Luciferase sequence was prepared by inserting the NotI-recognized sequence into pGL3 Luciferase Reporter Vectors (Promega) HindIII at the NcoI site, followed by digestion with NotI and XbaI. The sequences of PDK4, and HMGCS2 were amplified from murine cDNA using Q5 Hot Start High-Fidelity DNA Polymerase. pAAV9:cTNT::3Flag-hYAP (S127A) was a gift from William Pu (Addgene plasmid # 86558; http://n2t. net/addgene:86558; RRID:Addgene_86558) [20]. The oligo DNA sequences used for linker ligation and PCR are listed in

**Table 2. Primers used for making insert sequences.**

| target genes | | oligonucleotide sequences (5' → 3') |
|---|---|---|
| HindIII-NotI-NcoI linker | sense | AGCTTGCGGCCGCC |
| | antisense | CATGGGCGGCCGCA |
| AgeI-cTNT promoter | forward | AAAAACCGGTAATTCGCCCTTACGGGCCCC |
| | reverse | CCAAACCGGTTAGAGCTTCGGGGATCGTCC |
| EcoRI-Pdk4 | forward | AAAAGAATTCATGAAGGCAGCCCGCTTC |
| XbaI-Pdk4 | reverse | AAACTCTAGATTCACACTGCCAGCTTCTCC |
| BamHI-Hmgcs2 | forward | AAAAGGATCCATGCAGCGGCTTTTGGCTCC |
| HpaI-Hmgcs2 | reverse | CCCAGTTAACTTAGACGGGACACCGGGCAT |

Table 2. The shRNA expressing plasmids for gene knockdown were purchased from Sigma-Aldrich. PDK4 and HMGCS2 were silenced with TRCN0000023899 and TRCN0000075832, respectively. SHC001 was used for control shRNA.

### Lentivirus production and infection

Lentiviral vectors were generated as described previously [19]. Briefly, 5.2 µg pNHP, 2.1 µg pVSV, 0.5 µg pCEP, and 2.6 µg effector gene plasmid (pCSII or pLKO.1) were transfected to Lenti-X 293T cell (Takara Bio) using 60 µg PEI: Polyethylenimine "Max" (Polysciences) per 10 cm dish. 8–14 h after transfection, the medium was changed to DMEM GlutaMAX (Thermo Fisher Scientific) containing 10% FBS and cultured for approximately 36 h. To concentrate the lentivirus, the culture supernatant was centrifuged at 95000 × g for 2 h using XL-100K (Beckman Coulter) and SW28 (Beckman Coulter) centrifuges, followed by resuspension in PBS. Lentiviral RNA was extracted using NucleoSpin RNA virus (Takara Bio), and the titer was measured using a Lenti-X qRT-PCR Titration Kit (Takara Bio) according to the manufacturer's protocol. Lentiviral vectors were infected with NRCMs at 2000 VP/cell for 48 h.

### Statistical analysis

All statistical analyses were performed using Statcel Ver.4 (The Publisher OMS). Statistical differences between groups were determined using the Student's $t$-test, Dunnett test, or Tukey-Kramer test. Post hoc tests were performed only if the F-values analyzed using one-way ANOVA were significant. $p < 0.05$ was considered as significant.

## Results

### Fatty acids increase lipid metabolizing factors and arrest cell cycle progression in NRCMs

We investigated the effects of fatty acids on the expression of lipid metabolizing factors in NRCMs using the FA, the mixture of palmitic acid and oleic acid. Palmitic acid is one of the most abundant saturated fatty acids in mother milk. Oleic acid is also contained in mother milk plentifully and attenuates lipotoxicity induced by palmitic acid [21,22]. The FA increased Cd36, Cpt1b, Lcad, and Mcad mRNA levels in a dose-dependent manner, whereas glucose transporter 4 (Glut4) and hexokinase 2 (Hk2) mRNA levels were decreased (Fig 1A). Moreover, FA upregulated Pdk4 mRNA and protein (Fig 1A and S1 Fig), which inhibited pyruvate dehydrogenase complex to shift from glycolysis to β-oxidation, suggesting that FA activated β-oxidation instead of glycolysis even in high glucose conditions. In addition, FA also increased acetyl-CoA but did not affect ATP production (Fig 1B and 1C). We considered that excess acetyl-CoA was likely to be consumed for metabolic processes other than ATP production, such as ketogenesis. Although FA remarkably upregulated Hmgcs2 mRNA and protein, FA did not increase β-hydroxybutyrate as well as Hmgcl or Bdh1 mRNA (Fig 1D, 1E, and S1 Fig). These results show that the addition of FA upregulated several enzymes of β-oxidation and a ketogenic factor without ATP or ketone body production.

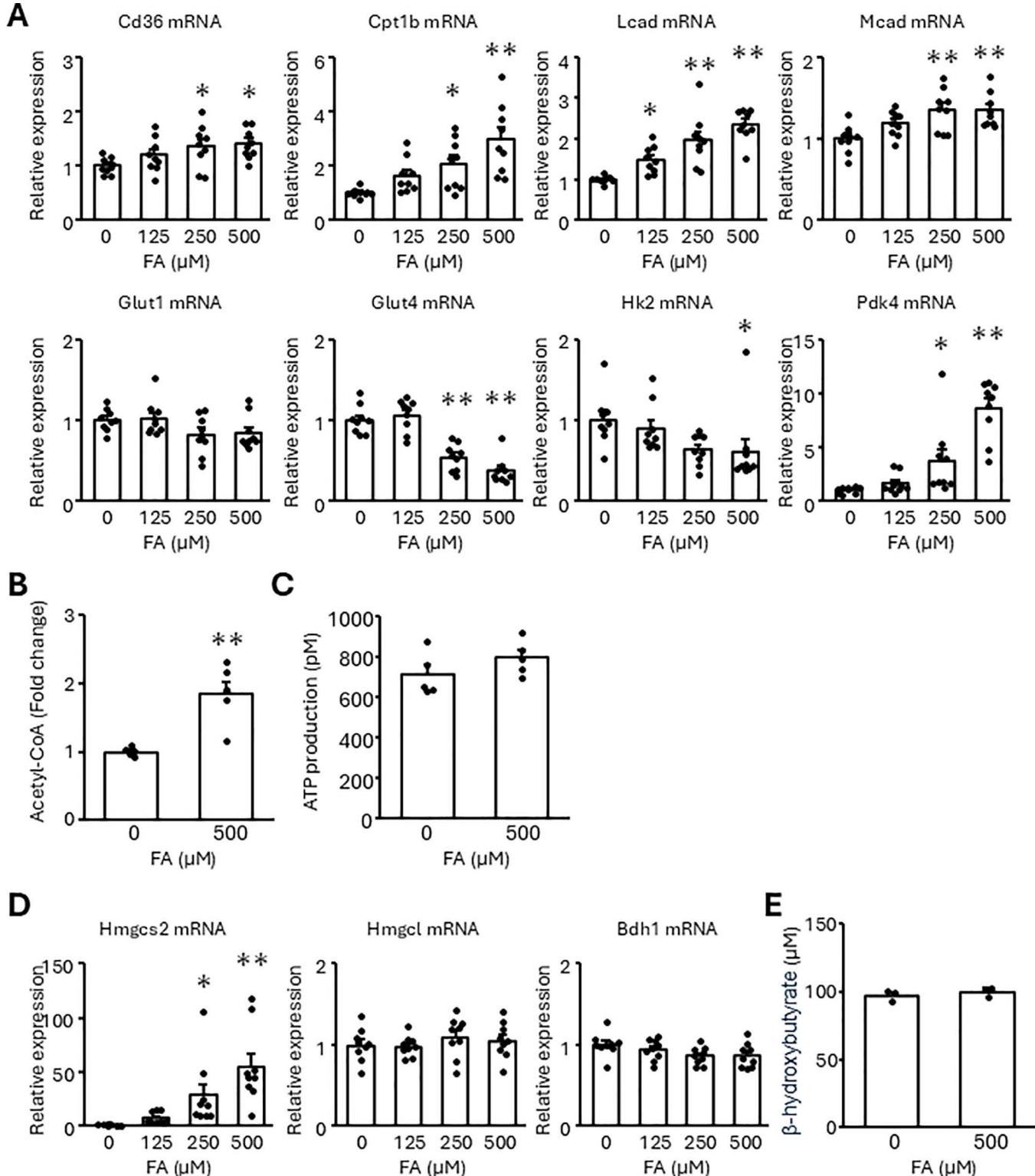

**Fig 1. FA treatment upregulates the expression of fatty acid metabolism-related genes without ATP production in NRCMs.** NRCMs were treated with the indicated concentrations of the FA for 24 h. (A, D) Transcript expression was measured using real-time RT-PCR. (B, C, E) Amounts of acetyl-CoA, ATP, and β-hydroxybutyrate were quantified. Results are shown as the mean ± SEM (A, D: n = 9, B, C: n = 5, E: n = 3). *p < 0.05, **p < 0.01 vs. 0 µM FA by Dunnett test (A, D) and Student's *t*-test (B, C, E).

Next, we examined whether the FA affected the proliferation of NRCMs using immunofluorescence microscopic analysis with an antibody against Ki67, a cell cycle marker. Under basal conditions, approximately 30% of NRCMs expressed Ki67, whereas treatment with the FA reduced the ratio of Ki67-positive NRCMs in a dose-dependent manner (Fig 2A and 2B). Adding 500 µM FA also decreased the proportion of NRCMs which expressed phospho-Histone H3, another cell cycle marker, by 75% (Fig 2C and 2D). Moreover, the intercellular expression of Aurora B was suppressed with FA treatment, suggesting the repression of cytokinesis (Fig 2E and 2F). Consistently, cell counting assay using WST-8 showed fewer NRCMs in 500 µM FA containing condition than in FA free condition (Fig 2G). These results indicated that fatty acid induced cell cycle arrest.

## FA disturbs NRCM regeneration through PPARδ

Fatty acid metabolism is positively regulated by transcriptional factors, including PPARα, PPARγ, and PPARδ [23]. We confirmed which PPAR subtype was mainly involved in fatty acid metabolism and cell cycle arrest in NRCMs. We treated NRCMs with fenofibrate, pioglitazone, and GW501516, agonists of PPARα, PPARγ, and PPARδ, respectively, and found that GW501516 upregulated the expression of β-oxidation related enzymes most remarkably (Fig 3A). The expression of Pdk4 and Hmgcs2 was also significantly increased in response to GW501516 treatment (Fig 3A and S2 Fig A, B). On the other hand, unlike the FA, none of the PPAR agonists affected Glut4 or Hk2 mRNA. Notably, GW501516 reduced the proportion of Ki67-positive NRCMs and that of intercellular Aurora B-positive NRCMs (Fig 3B, 3C, and S2 Fig C, D).

Next, we pretreated NRCMs with GW6471, T0070907, and GSK3787, antagonists of PPARα, PPARγ, and PPARδ, respectively, and examined their effects on fatty acid utilization and cell proliferation in response to FA treatment. Although GSK3787 suppressed the FA-induced increase of β-oxidation factors, including Pdk4 and Hmgcs2, either GW6471 or T0070907 had little or no influence (Fig 4A and S3 Fig A, B). Furthermore, GSK3787, but not the other inhibitors, completely recovered the reduction of Ki67-positive NRCMs in the presence of the FA (Fig 4B and 4C). The reduction of intercellular Aurora B-positive cells was also regained with GSK3787 (S3 Fig C and D). Incidentally, GSK3787 did not regain the reduction of Glut4 or Hk2 expression by FA. These results suggest that the cell cycle arrest resulted from the promotion of fatty acid metabolism mediated through PPARδ rather than the suppression of glycolysis in NRCMs.

## FA and PPARδ activation promotes NRCM maturation

Cardiomyocyte maturation is closely associated with cell cycle arrest. When zebrafish and newt repair the damaged heart, cardiomyocytes are dedifferentiated into immatured cells to reenter cell cycle [24,25]. Cardiomyocytes have two types of myosin heavy chain, namely Myh6 and Myh7. Since immature fetal and neonatal cardiomyocytes mainly express Myh7 and adult cardiomyocytes express Myh6 [26,27], the ratio between Myh7 and Myh6 (Myh7/Myh6) represents cardiomyocytes immature level. FA and GW501516 decreased Myh7 mRNA and Myh7/Myh6 ratio, and GSK3787 suppressed these reductions by FA (Fig 5), suggesting FA and PPARδ activation promotes NRCM maturation. These results correspond to the study reported by Wickramasinghe et al. to show that human iPS cell-derived cardiomyocytes were induced sarcomere maturation and binucleation by activating PPARδ [28].

## Oxidative stress and DNA damage have little effect on FA-induced cell cycle arrest

Oxidative stress disturbs cardiomyocyte proliferation because of damaging DNA [29,30]. A fatty acid challenge has previously been reported to increase oxidative stress in cardiomyocytes [31]. Consistently, the FA increased ROS levels, as analyzed using the CellROX Green assay (S4 Fig A). However, GW501516 did not increase ROS, or GSK3787 did not suppress FA-mediated ROS induction (S4 Fig B and C). γH2AX is known as a damaged DNA marker. Importantly, FA and GW501516 reduced the proportion of γH2AX-positive NRCMs and γH2AX protein level (S5 Fig). These results indicate that the PPARδ pathway-regulated cardiomyocyte cell cycle arrest was not mediated by ROS or DNA damage.

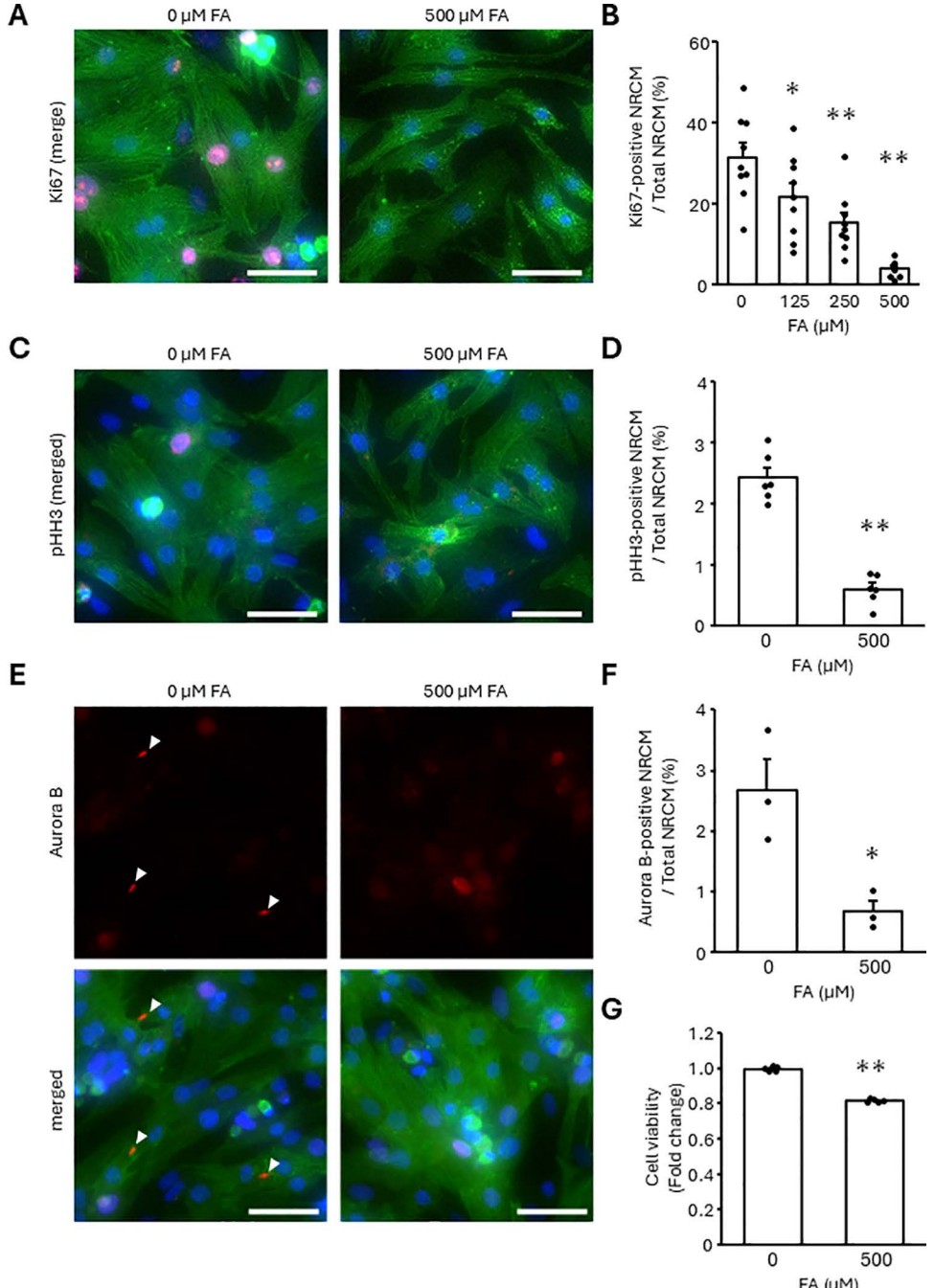

**Fig 2. FA suppressed NRCM proliferation.** NRCMs were treated with the indicated concentrations of the FA for 24 h (A-F) or 48 h (G). (A-F) The proportion of Ki67, phospho-Histone H3 (pHH3), or Aurora B-positive NRCMs was analyzed using immunostaining. Cells were stained with anti-Ki67, pHH3, or Aurora B-antibodies (red). Cardiomyocytes and nuclei were labeled with anti-α-actinin antibody (green) and DAPI (blue), respectively. (A, C, E) Representative images. (B, D, F) Quantitative data. The bars indicate 100 μm. Arrowheads indicate Aurora B. (G) The amount of NRCMs was analyzed by cell counting assay using WST-8. Results are shown as the mean ± SEM (B: n = 9, D: n = 6, F: n = 3, G: n = 5). *$p < 0.05$, **$p < 0.01$ vs. 0 μM FA by Dunnett test (B) and Student's *t*-test (D, F, G).

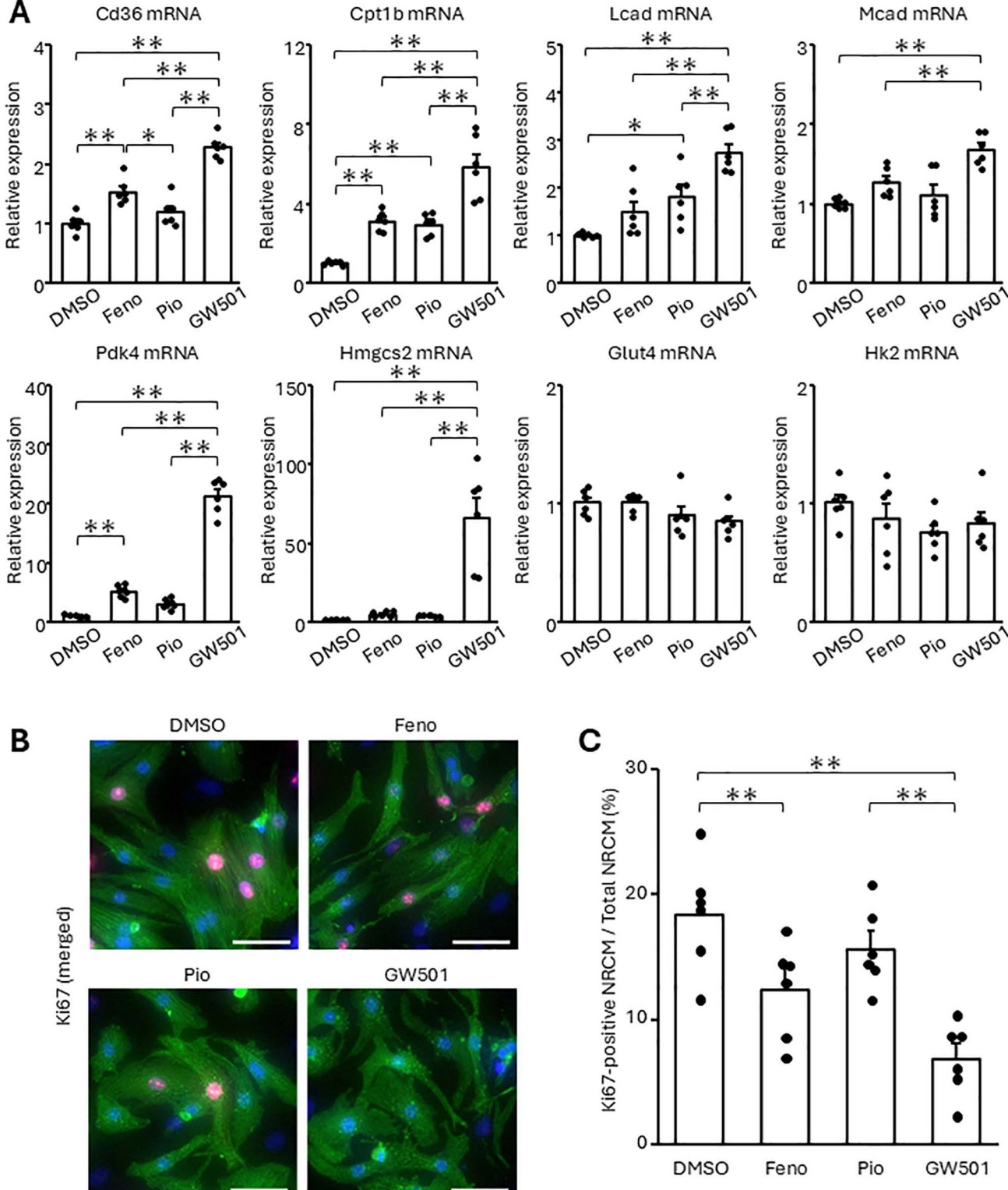

**Fig 3. The activation of PPAR δ upregulates fatty acid metabolism-related factors and suppresses NRCM proliferation.** NRCMs were treated with fenofibrate (Feno), pioglitazone (Pio), or GW501516 (GW501) at 10 μM for 24 h. (A) Transcript expression was measured using real-time RT-PCR. (B, C) The proportion of Ki67-positive NRCMs was analyzed using immunostaining. Cells were stained with an anti-Ki67 antibody (red). Cardiomyocytes

and nuclei were labeled with anti-α-actinin antibody (green) and DAPI (blue), respectively. (B) Representative images. (C) Quantitative data. The bars indicate 100 μm. Results are shown as the mean ± SEM (n = 6). *$p < 0.05$, **$p < 0.01$ by Tukey-Kramer test.

### The overexpression of PDK4 or HMGCS2 suppresses NRCM proliferation

Since FA or GW501516 increased the expressions of PDK4 and HMGCS2 more remarkably than that of β-oxidation-related factors, we investigated whether PDK4 and HMGCS2 were contributed to cell cycle arrest directly. Cardiomyocyte-specific lentiviral vectors were generated using a cTNT promoter. Infection with the Venus-expressing lentiviral vector (LV-Venus) showed that the vector was expressed in more than 90% of the NRCMs (S6 Fig). Importantly, transfection with LV-PDK4 or LV-HMGCS2 did not significantly affect the expression of β-oxidation factors (Fig 6A, and S7 Fig A). Notably, both LV-PDK4 and LV-HMGCS2 reduced the proportion of Ki67-positive NRCMs compared to cardiomyocytes infected with LV-Luciferase (Fig 6B and 6C). As opposed to the stimulation with FA or GW501516, PDK4 overexpression did not decreased, rather increased, Myh7/Myh6 ratio and HMGCS2 overexpression (S7 Fig A). Moreover, there were few Aurora B-positive NRCMs regardless of lentiviral vector (S7 Fig B).

### Knockdown of PDK4 or HMGCS2 regained NRCM cell cycle in FA-rich condition

To suppress PDK4 or HMGCS2 expression, we prepared lentiviral vectors to express shRNA targeting PDK4 or HMGCS2 (LV-shPDK4 or LV-shHMGCS2). In FA free condition, the further suppression of PDK4 or HMGCS2 was not observed in either LV-shPDK4- or LV-shHMGCS2-transfected NRCMs, compared to NRCMs infected with a control shRNA expressing lentiviral vector (LV-shControl). However, in FA containing condition, LV-shPDK4 reduced not only PDK4 but also HMGCS2. LV-shHMGCS2 also decreased both HMGCS and PDK4 (Fig 7A, 7B, and S8 Fig). Furthermore, both LV-shPDK4 and shHMGCS2 regained Ki67-positive NRCMs suppressed by FA treatment (Fig 7C and 7D). Since LV-shPDK4 suppressed FA-induced CD36, Cpt1b, and Mcad mRNA, LV-shHMGCS2 only suppressed CD36 (S8 Fig). These results suggest that PDK4 is involved in FA/PPARδ/β-oxidation pathway, while HMGCS2 does not belong to the pathway.

## Discussion

In the present study, we demonstrated that free fatty acids suppressed cardiomyocyte proliferation through the activation of PPARδ, accompanied by the induction of fatty acid metabolism-related enzymes, such as PDK4 and HMGCS2. In contrast, PPARδ blockade recovered the cell cycle arrest with the reduction of fatty acid metabolism. Overexpression of either PDK4 or HMGCS2 reduces cardiomyocyte regenerative capacity and suppressing these expressions regained cell cycle in lipid-rich condition. We revealed that fatty acids induce PDK4 and HMGCS2 and suppress the proliferative activities of neonatal cardiomyocytes through PPARδ.

Here, we show that fatty acid treatment suppresses the proliferation of cultured cardiomyocytes. Consistently, neonatal mice fed fat-free milk exhibited delayed cell cycle arrest in cardiomyocytes [12]. Moreover, Pdk4 ablation promoted cardiomyocyte proliferation. Thus, the enhancement of fatty acid utilization interrupts cardiac proliferation both in vivo and in vitro. It should be noted that the overexpression of PDK4 or HMGCS2 suppressed cardiomyocyte proliferation without promoting β-oxidation. Treatment with the FA or GW501516 upregulated the expression of Pdk4 and Hmgcs2 transcripts more remarkably than treatment with Cd36, Cpt1b, Lcad, and Mcad. Moreover, the activation of PPARδ with GW501516 suppressed NRCM proliferation without the addition of β-oxidation substrates. Collectively, the metabolic step responsible for PDK4 and HMGCS2 is more critical than β-oxidation in the cell cycle exit process in fatty acid-treated neonatal cardiomyocytes.

PDK4 inhibits the pyruvate dehydrogenase complex, which catalyzes the biochemical conversion of pyruvate to acetyl-CoA. HMGCS2 synthesizes ketone bodies from acetyl-CoA. Considering that both PDK4 and HMGCS2

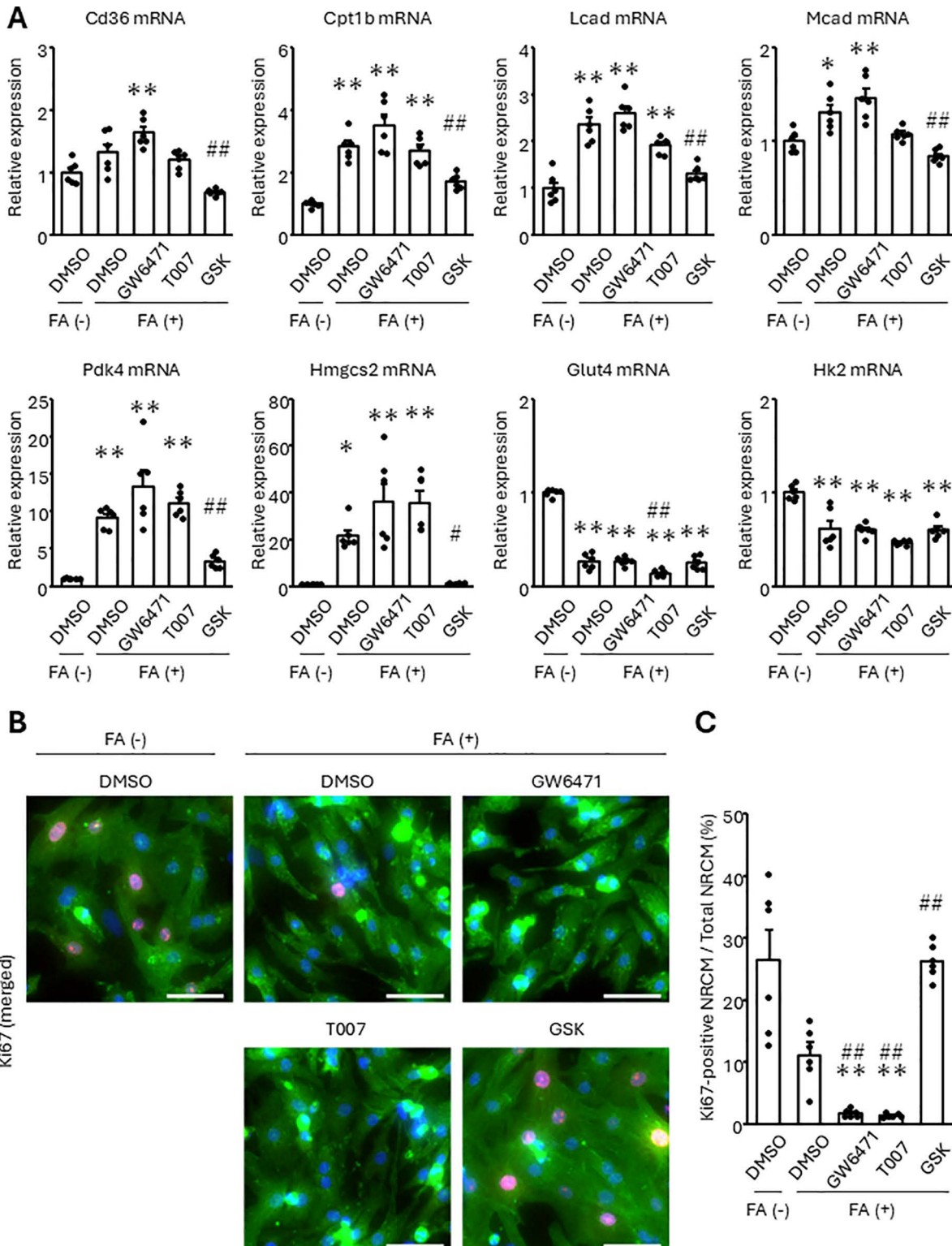

**Fig 4. Inhibition of PPAR δ represses the increase of fatty acid metabolism-related gene products and recovers the cell cycle arrest in response to FA treatment.** NRCMs were pretreated with GW6471, T0070907 (T007), or GSK3787 (GSK) at 10 µM for 24 h, followed by the treatment of 500 µM FA for 24 h. (A) Transcript expression was measured using real-time RT-PCR. (B, C) The proportion of Ki67-positive NRCMs was analyzed

using immunostaining. Cells were stained with an anti-Ki67 antibody (red). Cardiomyocytes and nuclei were labeled with anti-α-actinin antibody (green) and DAPI (blue), respectively. (B) Representative images. (C) Quantitative data. The bars indicate 100 μm. Results are shown as mean ± SEM. $*p < 0.05$, $**p < 0.01$ vs. FA (-), DMSO and $\#p < 0.05$, $\#\#p < 0.01$ vs. FA (+), DMSO by Tukey-Kramer test.

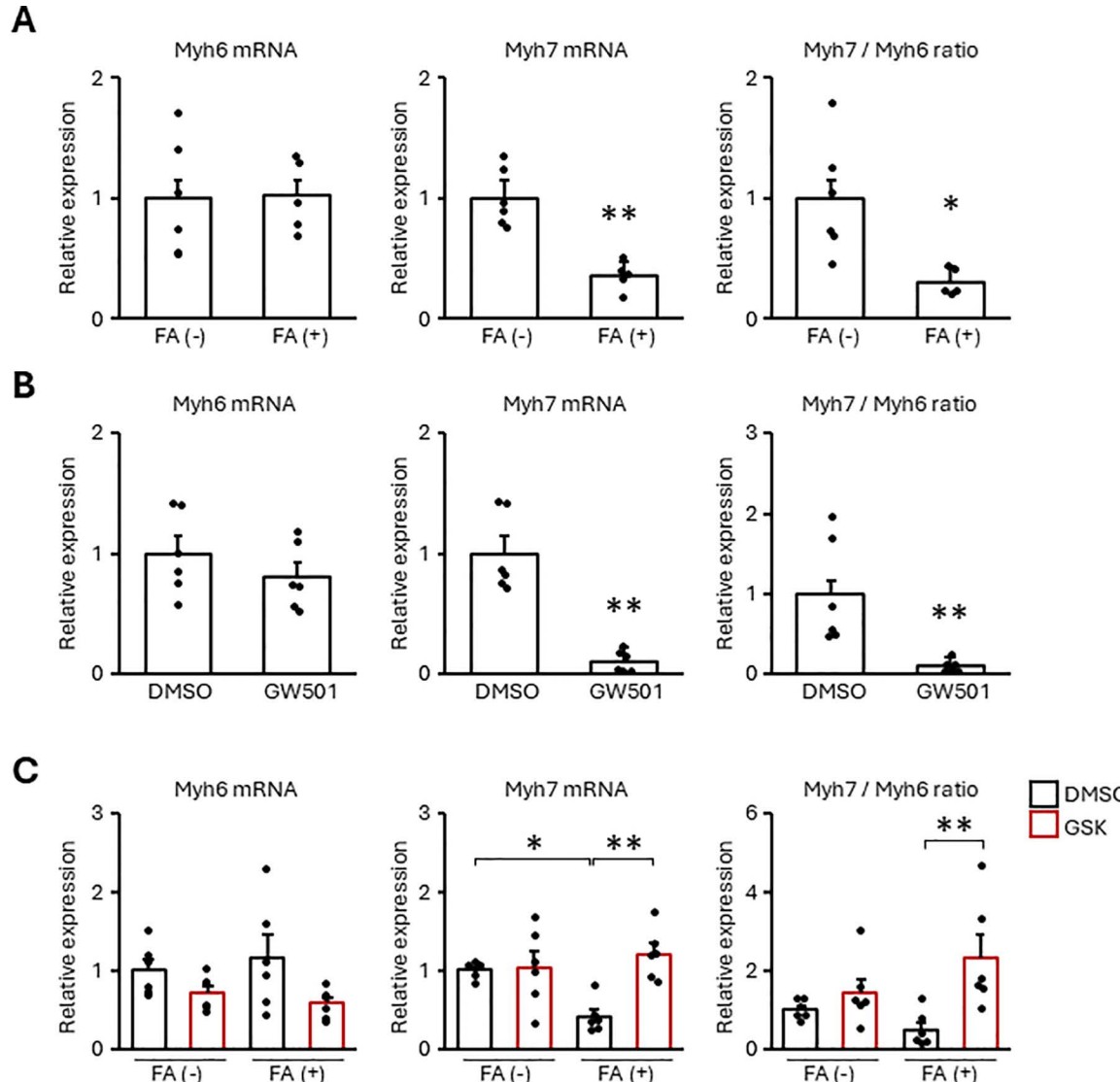

**Fig 5. FA promoted NRCM maturation through PPAR δ.** NRCMs were treated with 500 μM FA (A) or 10 μM GW501516 (GW501) (B) for 48 h. Otherwise, NRCMs were pretreated with GSK3787 (GSK) at 10 μM for 24 h, followed by the treatment of 500 μM FA for 48 h (C). The expression of the transcripts was measured by real-time RT-PCR. Results are shown as the mean ± SEM (A: n = 5-6, B, C: n = 6). $*p < 0.05$, $**p < 0.01$ vs. 0 μM FA by Student's $t$-test (A, B) and Tukey-Kramer test (C).

decreased acetyl-CoA levels, the reduction in acetyl-CoA suppressed cardiomyocyte proliferation. Acetyl-CoA is used in a wide range of biological pathways, such as the tricarbonyl cycle, ketone body formation, and protein acetylation [32]. We demonstrated that FA increased acetyl-CoA but did not alter the amount of ATP or β-hydroxybutyrate,

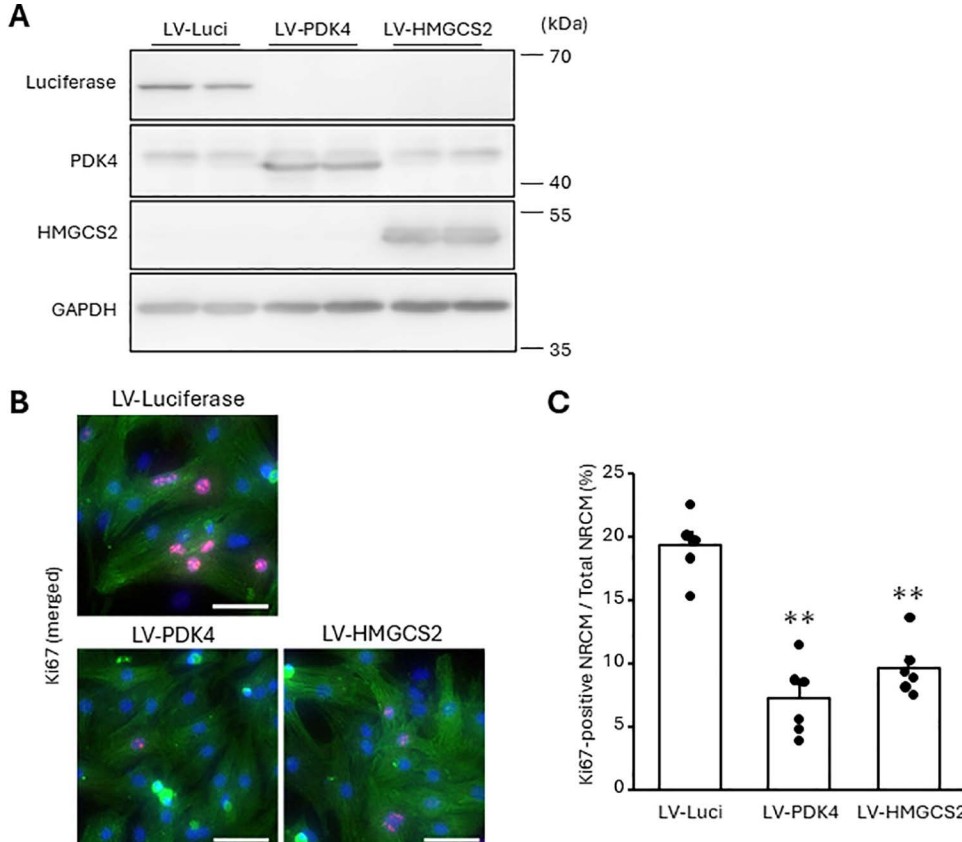

**Fig 6. The overexpression of PDK4 or HMGCS2 suppresses NRCM proliferation.** The NRCMs were infected with the indicated lentiviral vectors for 72 h. (A) Protein expression was measured using western blotting with anti-PDK4 and anti-HMGCS2 antibodies. Representative images are shown. (B, C) The proportion of Ki67-positive NRCMs was analyzed using immunostaining. Cells were stained with an anti-Ki67 antibody (red). Cardiomyocytes and nuclei were labeled with anti-α-actinin antibody (green) and DAPI (blue), respectively. (B) Representative images. (C) Quantitative data. The bars indicate 100 μm. Results are shown as mean ± SEM (n = 6). **$p < 0.01$ vs. LV-Luciferase by Dunnet test.

suggesting that fatty acid metabolism had little effect on the tricarbonyl cycle or ketone body formation. Interestingly, a previous study demonstrated that hypoxia-induced mitochondrial uncoupling protein 2 (UCP2) upregulation promoted the cardiomyocyte cell cycle associated with acetyl-CoA generation and histone H3 acetylation [33]. Thus, further efforts are needed to investigate whether fatty acid metabolism decreases acetyl-CoA and histone acetylation levels.

We demonstrated that inhibiting PPARα and PPARγ rather decreased Ki67-positive NRCMs in fat-rich condition. We also showed that both PPARα and PPARγ inhibitor slightly increased the upregulation of Pdk4 and Hmgcs2 induced by FA stimulation. Previous studies revealed that both PPARα inhibitor and PPARγ inhibitor induced cell cycle arrest [34,35] and that the activation of PPARα prevents from fatty acid-induced apoptosis in NRCMs [36]. Therefore, PPARα and PPARγ are likely to have a protective effect on cardiomyocyte proliferation against fatty acid.

In this study, PPARδ activation by GW501516 inhibited the NRCM cell cycle; however, a previous study demonstrated that PPARδ promoted the proliferation by using carbacyclin [37]. It is possible that the difference in the outcomes between the two studies resulted from the difference of PPARδ agonists as a limitation of pharmacological specificity. In this context, by using the PPARδ antagonist, we confirmed that the blockade of PPARδ recovered the suppression of NRCM proliferation.

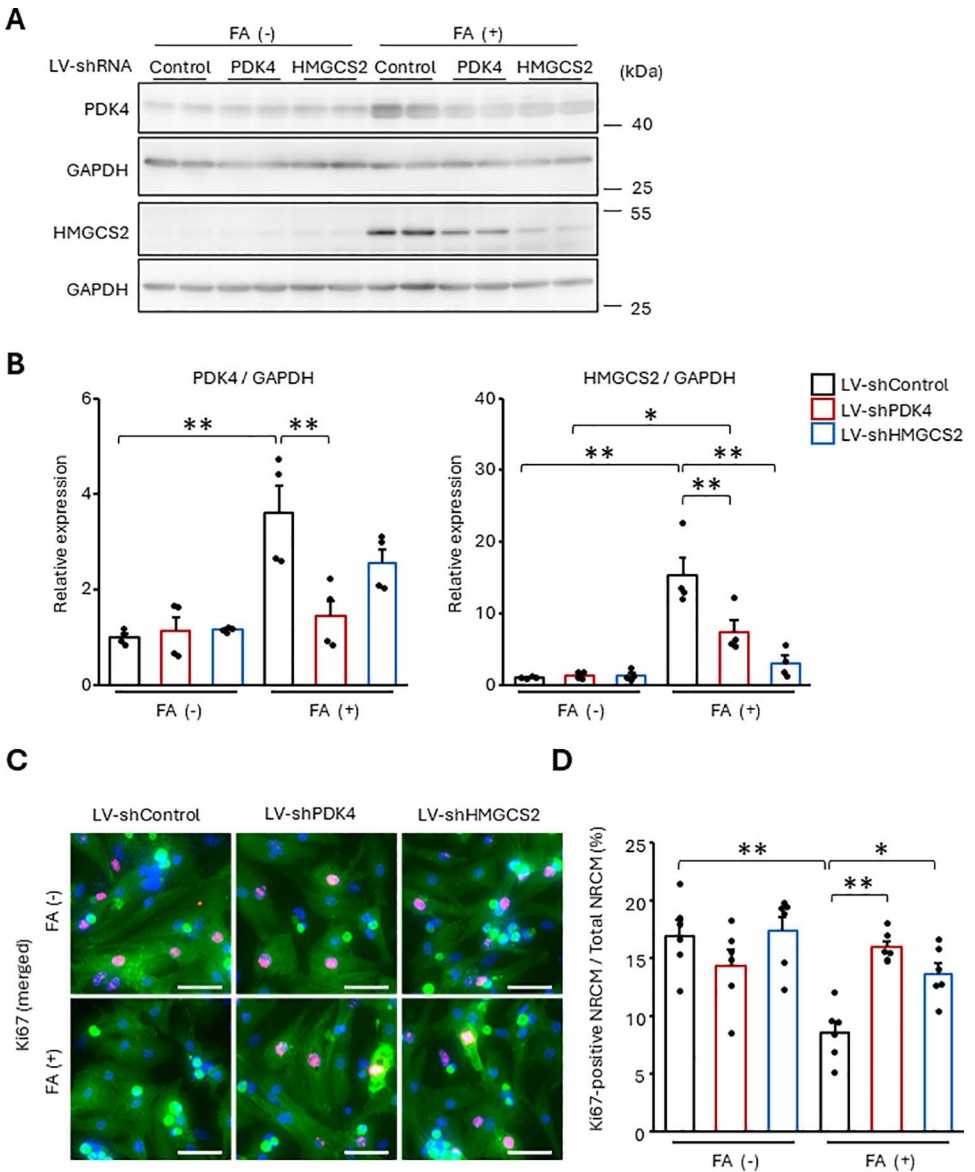

**Fig 7. The suppression of PDK4 or HMGCS2 abrogates FA-induced cell cycle arrest.** The NRCMs were infected with the indicated lentiviral vectors for 48 h, followed by the stimulation of 500 µM FA. (A, B) Protein expression was measured using western blotting with anti-PDK4 and anti-HMGCS2 antibodies. (C, D) The proportion of Ki67-positive NRCMs was analyzed using immunostaining. Cells were stained with an anti-Ki67 antibody (red). Cardiomyocytes and nuclei were labeled with anti-α-actinin antibody (green) and DAPI (blue), respectively. (A, C) Representative images. (B, D) Quantitative data. The bars indicate 100 µm. Results are shown as mean ± SEM (B: n = 4, D: n = 6). *$p < 0.05$, **$p < 0.01$ by Tukey-Kramer test.

Although FA and GW501516 suppressed NRCM cytokinesis, analyzed by the intercellular staining of Aurora B, overexpressing PDK4 or HMGCS2 did not affect cell division. Moreover, unlike FA and GW501516, PDK4 and HMGCS2 overexpression did not decreased Myh7/Myh6 ratio, suggesting these overexpressions did not induce NRCM maturation. Actually, Cheng et al. reports HMGCS2 induced adult cardiomyocytes dedifferentiation [15]. The signaling pathway of FA-induced NRCM maturation might not be identical to that of PDK4 and HMGCS2, though these two enzymes are downstream of FA.

Similar to our study, Wickramasinghe et al. demonstrated that PPARδ activation by GW0742 upregulated several lipid metabolism factors including Pdk4 and Hmgcs2 by RNA-sequence analysis using human iPS-derived cardiomyocytes [28], proposing that PPARδ activation is useful to make high-differentiated iPS-derived cardiomyocytes that exhibit enhanced mitochondrial respiration and matured sarcomere to constriction. Collectively, the overexpression of PDK4 or HMGCS2 is not likely to induce sarcomere maturation but to reinforce mitochondrial maturation, contributing to the metabolic differentiation of cardiomyocytes.

Puente et al. describes that oxidative stress induces DNA damage, resulting in cell cycle arrest in cardiomyocytes [29]. On the other hand, the present study showed that FA increased ROS in NRCMs but that DNA damage, estimated by the expression of γH2AX, was rather decreased with FA stimulation. The contradicts may be explained by the hypothesis that the activation of PPARδ attenuated DNA damage. In this context, Ding et al. demonstrated that PPARδ is essential to protect cardiomyocytes from oxidative stress [38]. Consistently, GW501516 actually reduced DNA damage without the gain of ROS level in the present study. Thus, ROS-induced DNA damage is unlikely to be involved in the interruption of NRCM proliferation induced with FA.

In conclusion, our data, presented here, demonstrated that PPARδ-mediated induction of PDK4 or HMGCS2 is sufficient for cell cycle arrest associated with metabolism rearrangement in neonatal cardiomyocytes. Furthermore, the inhibition of PPARδ contributes to the maintenance of cell cycle activity in lipid-rich conditions. PPARδ signaling could be a therapeutic target in myocardial regenerative therapy.

## Supporting information

**S1 Fig. FA treatment increases the protein expression of PDK4 and HMGCS2.** NRCMs were treated with 500 μM FA for 24 h. The protein expression of PDK4 and HMGCS2 was measured using western blotting with anti-PDK4 and anti-HMGCS2 antibodies. (A) Representative images. (B) Quantitative data. Data are shown as mean ± SEM (n = 6). **$p <$ 0.01 by Student's $t$-test.
(TIF)

**S2 Fig. The activation of PPARδ reduces NRCM cytokinesis with the increases of PDK4 and HMGCS2.** NRCMs were treated with fenofibrate (Feno), pioglitazone (Pio), or GW501516 (GW501) at 10 μM for 24 h. (A, B) Protein expression was measured using western blotting with anti-PDK4 and anti-HMGCS2 antibodies. (C, D) The proportion of Aurora B-positive NRCMs was analyzed by immunostaining. Cells were stained with an anti-Aurora B antibody (red). Cardiomyocytes and nuclei were labeled with anti-α-actinin antibody (green) and DAPI (blue), respectively. (A, C) Representative images. (B, D) Quantitative data. The bars indicate 100 μm. Arrowheads indicate Aurora B. Data are shown as mean ± SEM (A: n = 6, B: n = 3). *$p <$ 0.05, ** $p <$ 0.01 by Dunnett test (A) and Student's $t$-test (B).
(TIF)

**S3 Fig. The inhibition of PPARδ regained the suppression of NRCM cytokinesis by FA.** NRCMs were pretreated with GSK3787 (GSK) at 10 μM for 24 h, followed by the treatment of 500 μM FA for 24 h. (A, B) Protein expression was measured using western blotting with anti-PDK4 and anti-HMGCS2 antibodies. (C, D) The proportion of Aurora B-positive NRCMs was analyzed by immunostaining. Cells were stained with an anti-Aurora B antibody (red). Cardiomyocytes and nuclei were labeled with anti-α-actinin antibody (green) and DAPI (blue), respectively. (A, C) Representative images. (B, D) Quantitative data. The bars indicate 100 μm. Arrowheads indicate Aurora B. Results are shown as mean ± SEM (A: n = 7, B: n = 3). *$p <$ 0.05, **$p <$ 0.01 by Tukey-Kramer test.
(TIF)

**S4 Fig. Oxidative stress is not related to the loss of NRCM proliferative activity in response to FA.** (A, B) Reactive oxygen species (ROS) levels were measured using the fluorescence of CellROX Green 24 h after the stimulation

with 500 µM FA (A) or 10 µM GW501516 (GW501) (B) in NRCMs. Data are shown as mean ± SEM (n = 5). **$p < 0.01$ by Student's *t*-test. (C) Data are shown as mean ± SEM (n = 5). *$p < 0.05$, **$p < 0.01$ by Tukey-Kramer test. n.s. indicates no significance.
(TIF)

**S5 Fig. The stimulation with FA or GW501516 decreases DNA damage.** NRCMs were treated with 500 µM FA (A, B, E, F) or 10 µM GW501516 (GW501) (C, D, G, H) for 24 h. (A-D) The proportion of γH2AX-positive NRCMs was analyzed by immunostaining. Cells were stained with an anti-γH2AX antibody (red). Cardiomyocytes and nuclei were labeled with anti-α-actinin antibody (green) and DAPI (blue), respectively. The bars indicate 100 µm. (E-H) Protein expression was measured by western blotting with anti-γH2AX antibody. (A, C, E, G) Representative images. (B, D, F, H) Quantitative data. Data are shown as mean ± SEM (A, B: n = 3, C, D: n = 6). *$p < 0.05$, **$p < 0.01$ by Student's *t*-test.
(TIF)

**S6 Fig. NRCMs express objective gene by infecting cardiomyocyte specific lentiviral vectors.** The NRCMs were infected with the indicated lentiviral vectors for 72 h. (A) Transfection efficacy was estimated using Venus expression (green). Cardiomyocytes and nuclei were labeled with anti-α-actinin antibody (red) and DAPI (blue), respectively. Representative images are shown. The bars indicate 100 µm.
(TIF)

**S7 Fig. Neither PDK4 nor HMGCS2 overexpression increases β-oxidation related enzymes.** (A) NRCMs were infected with the indicated lentiviral vectors for 48 h, followed by the treatment of 500 µM FA for 24 h. Transcript expression was measured using real-time RT-PCR. (B) The proportion of Aurora B-positive NRCMs was analyzed by immunostaining. Cells were stained with anti-Aurora B antibody (red). Cardiomyocytes and nuclei were labeled with anti-α-actinin antibody (green) and DAPI (blue), respectively. The bars indicate 100 µm. Results are shown as the mean ± SEM (n = 6). **$p < 0.01$ vs. LV-Luciferase by Dunnet test.
(TIF)

**S8 Fig. Silencing PDK4, but not HMGCS2, reduces FA-upregulated β-oxidation enzymes.** The NRCMs were infected with the indicated lentiviral vectors for 48 h, followed by the stimulation of 500 µM FA. The expression of transcripts was measured by real-time RT-PCR. Results are shown as mean ± SEM (n = 4). *$p < 0.05$, **$p < 0.01$ by Tukey-Kramer test.
(TIF)

**S1 raw images. Original images for blots.**
(PDF)

## Author contributions

**Conceptualization:** Shota Tanaka.

**Data curation:** Shota Tanaka, Akane Hirota.

**Formal analysis:** Shota Tanaka, Akane Hirota.

**Funding acquisition:** Shota Tanaka, Yasushi Fujio.

**Project administration:** Shota Tanaka.

**Writing – original draft:** Shota Tanaka.

**Writing – review & editing:** Akane Hirota, Yoshiaki Okada, Masanori Obana, Yasushi Fujio.

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
