## [Decision Letter · Decision Letter 0]

21 Aug 2024

PONE-D-24-32166Fatty acid metabolism suppresses cardiomyocyte proliferation by increasing PDK4 and HMGCS2 expression through PPARδPLOS ONE

Dear Dr. Tanaka,

Thank you for submitting your manuscript to PLOS ONE. After careful consideration, we feel that it has merit but does not fully meet PLOS ONE’s publication criteria as it currently stands. Therefore, we invite you to submit a revised version of the manuscript that addresses the points raised during the review process.

As the reviewer noted, it is still unclear whether the number of cardiomyocytes is actually reduced under these conditions, so this study would be better served by a more in-depth assessment of cardiomyocyte proliferation and maturation.

We look forward to receiving your revised manuscript.

Kind regards,

Koh Ono, M.D., Ph.D.

Academic Editor

PLOS ONE

Journal Requirements:

"This study is partially supported by MEXT/JSPS KAKENHI Grants 20K22707 to ST, 22K15277 to ST. This research was also supported by Basis for Supporting Innovative Drug Discovery and Life Science Research (BINDS) from AMED under grant numbers JP23ama121052 and JP23ama121054."

"The authors have no conflicts of interest associated with this manuscript."

Additional Editor Comments:

As the reviewers noted, it is still unclear whether the number of cardiomyocytes is actually reduced under these conditions, so this study would be better served by a more in-depth assessment of cardiomyocyte proliferation and maturation.

Reviewers' comments:

Reviewer's Responses to Questions

**Comments to the Author**

1. Is the manuscript technically sound, and do the data support the conclusions?

Reviewer #1: Partly

Reviewer #2: Partly

2. Has the statistical analysis been performed appropriately and rigorously? 

Reviewer #1: Yes

Reviewer #2: Yes

3. Have the authors made all data underlying the findings in their manuscript fully available?

Reviewer #1: Yes

Reviewer #2: Yes

4. Is the manuscript presented in an intelligible fashion and written in standard English?

Reviewer #1: Yes

Reviewer #2: Yes

5. Review Comments to the Author

Reviewer #1: Although cardiomyocytes lose proliferating ability after birth quickly, the mechanism is not fully understood. Growing evidences show that one of the mechanisms is metabolic shift from glycolysis to fatty acid (FA) oxidation in cardiomyocytes after birth. Others also demonstrated that ketone derived from FA plays pivotal role in cell cycling arrest in cardiomyocytes. In this article, Tanaka et al. found that FA stimulation for neonatal rat cardiomyocytes suppressed Ki67 expression which is a positive marker of cell cycling. Since FA stimulation induced striking upregulation of Pdk4, a key regulator for activation of beta-oxidation, and Hmgsc2, a positive regulator for ketone production, and FA metabolism is positively regulated by PPARs, using PPAR isoform specific agonists and antagonists Tanaka et al. identified that PPARdelta is a major transcription factor that induces Pdk4 and Hmgcs2 transcripts and that the upregulations resulted in cell cycle arrest. The authors conclude that FA metabolism suppresses rat neonatal cardiomyocyte proliferation by upregulation of Pdk4 and Hmgcs2 regulated by PPARdelta.

Given that many studies using genetically modified mice and iPS cell-derived cardiomyocytes demonstrated that FA metabolism suppresses cardiomyocyte proliferation and induces the maturation, it would be better to assess cardiomyocytes proliferation and maturation more deeply in this in vitro only study. Especially the authors claimed that FA suppresses neonatal cardiomyocyte proliferation. However it is still unknown if the number of cardiomyocytes is actually decreased under this condition. Major concerns are listed below followed by minor concerns.

1. Fig. 1 and 3: The authors should show the protein upregulations of PDK4 and HMGCS2 in neonatal cardiomyocytes stimulated with FA.

2. Fig. 1: The authors described that excess acetyl-CoA is likely to be consumed for the metabolic processes other than ATP production, such as ketogenesis. Did the authors assess acetyl-CoA levels in cardiomyocytes with FA? Also, ketone and/or the metabolites are increased in the myocytes stimulated with FA?

3. Fig. 2: The readers would wonder if neonatal cardiomyocytes are actually proliferating at basal condition. Ki67 and pHH3 are used for cell-cycling indicators. But it is still unclear if cytokinesis and karyokinesis are happening in the cardiomyocytes since mouse cardiomyocyte is an endomitotic cell. To clarify this concern, it would better to assess expression levels of Aurora kinase B, Anillin, and Prc1 at transcript and protein levels. Immunocytochemistry also can detect the expressions of these proteins. Furthermore, the number of cells should be assessed to see the proliferation.

4. Fig. 2: What is the consequence in Ki67 negative neonatal cardiomyocytes stimulated with FA? Do they maturate quickly? Evaluation of binucleation and sarcomere component isoform switch will answer this question.

5. Fig. 3 and 4: Again many readers will have a question about proliferation and maturation in neonatal cardiomyocytes with PPARdelta agonist and antagonist.

6. Fig. 5B and C: If cardiomyocytes are not stimulated with FA in Fig. 5B, can ROS be increased without beta-oxidation substrates? Furthermore, since there is a difference in ROS levels of cardiomyocytes stimulated with FA alone between Fig. 5A and 5C, it is still inconclusive about that GSK did not suppress FA-induced ROS production in Fig. 5C. How can the readers interpret this inconsistency?

7. Can shRNA-PDK4 and/or shRNA-HMGCS2 rescue the cell cycle arrest by FA stimulation or PPARdelta agonist?

8. Fig. 6: Did overexpressions of PDK4 and HMGCS2 change proliferation and maturation of neonatal cardiomyocytes after cell-cycling arrest?

Minor

1: Since neonatal cardiomyocytes and adult cardiomyocytes are completely different, it is recommended to use the word “neonatal cardiomyocytes” in the title.

2: Given that there are multiple controls, it is better to use one-way ANOVA with post hoc test in Fig. 4.

Reviewer #2: Summary and overall impression

The authors sought to identify fatty acid metabolism contribution to cell cycle regulation in neonatal cardiomyocytes using neonatal rat cardiomyocytes. The finding is remarkable and interesting, but some concerns persist.

Evidence and examples

Major issues

1. Line 123 “fatty acid mixture (FA) containing palmitic acid, oleic acid, and L-carnitine at a 1:1:2 molar ratio”: What is this mixture trying to model? Please clarify the situation in human body that is mimicked by this mixture for cardiomyocytes beyond the abundance of each FA.

2. Line 247 “Besides, excess acetyl-CoA is likely to be consumed for the metabolic processes other than ATP production, such as ketogenesis.”: The logic here is obscure. Please clarify why the authors speculated as stated.

3. The inhibition of PDK4 and HMGCS2 is obscure whether to inhibit PPARδ -> β oxidation pathway or not. Please consider rescue experiments to show this.

4. Line 366: The FA stimulation is not directly shown to activate PPARδ in this manuscript. Please indicate if this has been shown previously.

5. If there is publicly available screening data using PPARs for cardiomyocytes, it might be interesting to see the difference between them to make the observed differences in this study.

6. PLOS authors have the option to publish the peer review history of their article (what does this mean? ). If published, this will include your full peer review and any attached files.

**Do you want your identity to be public for this peer review?** For information about this choice, including consent withdrawal, please see our Privacy Policy .

Reviewer #1: No

Reviewer #2: No

---

## [Author Response · Author response to Decision Letter 1]

31 Dec 2024

The Editor

PLOS ONE

Manuscript number: PONE-D-24-32166

Please find the revised manuscript entitled “Fatty acid metabolism suppresses neonatal cardiomyocyte proliferation by increasing PDK4 and HMGCS2 expression through PPARδ“ by Tanaka, S. et al. We revised the manuscript according to the reviewers’ comments. The responses to the comments are described as “comments for revision”. We thank the editors and reviewers for their comments, which we believe have improved our manuscript. We hope that this manuscript will be reconsidered for publication in PLOS ONE.

All authors have read this manuscript and agree to submit it to your journal. We haven’t submitted any results of this manuscript elsewhere.

This study is partially supported by MEXT/JSPS KAKENHI Grants 20K22707 to ST, 22K15277 to ST. This research was also supported by Basis for Supporting Innovative Drug Discovery and Life Science Research (BINDS) from AMED under grant numbers JP23ama121052 and JP23ama121054. There was no additional external funding received for this study. In addition, the authors have no conflicts of interest associated with this manuscript.

~ Comments for Revision ~

Reviewer #1

Comment #1: Fig. 1 and 3: The authors should show the protein upregulations of PDK4 and HMGCS2 in neonatal cardiomyocytes stimulated with FA.

Response: According to the reviewer’s comment, we measured PDK4 and HMGCS2 protein expressions and confirmed FA and GW501516 increased both expressions. These results are shown in S1 Fig and S2 Fig.

Comment #2: Fig. 1: The authors described that excess acetyl-CoA is likely to be consumed for the metabolic processes other than ATP production, such as ketogenesis. Did the authors assess acetyl-CoA levels in cardiomyocytes with FA? Also, ketone and/or the metabolites are increased in the myocytes stimulated with FA?

Response: According to the reviewer’s comment, we measured acetyl-CoA and β-hydroxybutyrate, a ketone body, in FA-treated cardiomyocytes. Although FA upregulated acetyl-CoA and Hmgcs2 level, β-hydroxybutyrate level was not altered by FA treatment. These data indicate the addition of FA upregulated several enzymes of β-oxidation and a ketogenic factor without ATP or ketone body production. These results are shown in Figure 1B and 1E.

Comment #3: Fig. 2: The readers would wonder if neonatal cardiomyocytes are actually proliferating at basal condition. Ki67 and pHH3 are used for cell-cycling indicators. But it is still unclear if cytokinesis and karyokinesis are happening in the cardiomyocytes since mouse cardiomyocyte is an endomitotic cell. To clarify this concern, it would better to assess expression levels of Aurora kinase B, Anillin, and Prc1 at transcript and protein levels. Immunocytochemistry also can detect the expressions of these proteins. Furthermore, the number of cells should be assessed to see the proliferation.

Response: We thank the reviewer for the important suggestion. As suggested by the reviewer, Aurora B, Anillin, and Prc1 mRNA were measured using real-time PCR and little alteration was observed by FA. Therefore, we analyzed NRCMs cytokinesis using immunostaining with an anti-Aurora B antibody and revealed that FA reduced intracellular Aurora B. Moreover, we quantified the number of NRCMs using the WST-8 assay and confirmed that there were fewer NRCMs in FA contained condition. These data indicate that cytokinesis ocurrs in NRCMs during proliferative process. Moreover, FA suppresses proliferation in NRCMs. These results are shown in Figure 2E, 2F, and 2G.

Comment 4: Fig. 2: What is the consequence in Ki67 negative neonatal cardiomyocytes stimulated with FA? Do they maturate quickly? Evaluation of binucleation and sarcomere component isoform switch will answer this question.

Response: As suggested by the reviewer, we measured the transcript expression of sarcomere components, such as Myh6 and Myh7, and revealed that FA suppressed Myh7 mRNA without the change of Myh6. In addition, the ratio of Myh7/Myh6 was also reduced after FA treatment, suggesting FA promotes NRCM maturation. The results were shown in Figure 5. We tried the experiment to evaluate NRCMs binucleation. However, we failed to analyze binucleation due to technical limitation: To check the number of nuclei, we seeded cells sparsely than usual, resulting in the severe damage of NRCMs in the presence of FA.

Comment #5: Fig. 3 and 4: Again many readers will have a question about proliferation and maturation in neonatal cardiomyocytes with PPARdelta agonist and antagonist.

Response: According to the reviewer’s comment, we measured Aurora B expression and Myh7 mRNA in NRCMs treated by PPARδ agonist- or antagonist to evaluate proliferation and maturation, respectively. A PPARδ agonist GW501516 reduced Aurora B-positive NRCMs, accompanied by the decreases expression of Myh7 mRNA. Moreover, a PPARδ inhibitor GSK3787 regained the reduction of Aurora B-positive NRCMs, accompanied by the increase in Myh7 mRNA. These results indicate that PPARδ activation promotes NRCM maturation. The results were shown in Figure 5, S2 Fig, and S3 Fig.

Comment #6: Fig. 5B and C: If cardiomyocytes are not stimulated with FA in Fig. 5B, can ROS be increased without beta-oxidation substrates? Furthermore, since there is a difference in ROS levels of cardiomyocytes stimulated with FA alone between Fig. 5A and 5C, it is still inconclusive about that GSK did not suppress FA-induced ROS production in Fig. 5C. How can the readers interpret this inconsistency?

Response: We apologize for the confusing presentation. In Fig. 5B (original), we examined whether a PPARδ agonist GW501516 increased ROS level without FA stimulation. The treatment with GW501516 only, without β-oxidation substrates, did not increase ROS. Furthermore, when the absolute value of fluorescence intensity was compared, there was no significant difference in the FA alone group between Fig. 5A and 5C (data not shown). In the experiments, we examined whether ROS-induced DNA damage was involved in NRCMs cell cycle arrest. Although FA increased ROS level in NRCMs, FA or GW501516 did not increase γH2AX, a damaged DNA marker. These data suggest that FA or GW501516 did not stop NRCMs proliferation through ROS-induced DNA damage. Since the effects of a PPARδ inhibitor GSK3787 on ROS level in NRCMs was obscure in Fig 5C (original), we consider that the results of ROS analysis were suitable for supportive data and original Figure 5 was transferred to S4 Fig. The results of γH2AX expression were shown in S5 Fig and the discussion was described at Line 512.

Comment #7: Can shRNA-PDK4 and/or shRNA-HMGCS2 rescue the cell cycle arrest by FA stimulation or PPARdelta agonist?

Response: As suggested by the reviewer, we suppressed PDK4 and HMGCS2 using shRNA expressing lentiviral vectors and measured Ki67-positive NRCMs after FA stimulation. Silencing either PDK4 or HMGCS2 recovered FA-induced NRCMs cell cycle arrest. The results were shown in Figure 7.

Comment #8: Fig. 6: Did overexpressions of PDK4 and HMGCS2 change proliferation and maturation of neonatal cardiomyocytes after cell-cycling arrest?

Response: We apologize to the reviewer for our lack of understanding. Overexpressing PDK4 or HMGCS2 might still stop cell cycle after cell cycle arrest. In addition, we analyzed proliferation and maturation in PDK4- or HMGCS2-overexpressing NRCMs. We revealed that Aurora B was not detected in Luciferase-overexpressing control NRCMs as well as PDK4- or HMGCS2-overexpressing cells after cell-cycling arrest. Moreover, PDK4 overexpression did not alter Myh7 mRNA and HMGCS2 overexpression rather increased Myh7. The results were shown in S7 Fig and possible reasons were discussed at Line 496.

Minor

Comment #1: Since neonatal cardiomyocytes and adult cardiomyocytes are completely different, it is recommended to use the word “neonatal cardiomyocytes” in the title.

Response: According to the reviewer’s comment, we change the title to “Fatty acid metabolism suppresses neonatal cardiomyocyte proliferation by increasing PDK4 and HMGCS2 expression through PPARδ“.

Comment #2: Given that there are multiple controls, it is better to use one-way ANOVA with post hoc test in Fig. 4.

Response: We thank the reviewer for the important suggestion. Based on your suggestion, we used one-way ANOVA followed by Tukey-Kramer test in Figure 4.

Reviewer #2:

Major issues

Comment #1: Line 123 “fatty acid mixture (FA) containing palmitic acid, oleic acid, and L-carnitine at a 1:1:2 molar ratio”: What is this mixture trying to model? Please clarify the situation in human body that is mimicked by this mixture for cardiomyocytes beyond the abundance of each FA.

Response: We thank the reviewer for the important suggestion. We used FA to mimic neonatal heart condition to be fed mother milk. Thus, we described in manuscript as follow:

To investigate the effects of fatty acids on NRCMs in neonatal model, we used an FA, which was the mixture of palmitic acid and oleic acid. Palmitic acid is one of the most abundant saturated fatty acids in mother milk. Oleic acid is also contained in mother milk plentifully and attenuates lipotoxicity induced by palmitic acid (21, 22).

Comment #2: Line 247 “Besides, excess acetyl-CoA is likely to be consumed for the metabolic processes other than ATP production, such as ketogenesis.”: The logic here is obscure. Please clarify why the authors speculated as stated.

Response: We apologize to the reviewer for our lack of reason. We confirmed that FA increased acetyl-CoA. The result was shown in Figure 1B.

Comment #3: The inhibition of PDK4 and HMGCS2 is obscure whether to inhibit PPARδ -> β oxidation pathway or not. Please consider rescue experiments to show this.

Response: According to the reviewer’s comment, we measured the transcript expressions of β-oxidation enzymes in either PDK4 or HMGCS2 silencing NRCMs. Suppressing PDK4 reduced FA-induced β-oxidation enzymes, including Cd36, Cpt1b, and Mcad, whereas suppressing HMGCS2 did not any except for Cd36 mRNA. These results indicate that the suppression of PDK4 inhibits PPARδ/β-oxidation pathway, while not that of HMGCS2.

Comment #4: Line 366: The FA stimulation is not directly shown to activate PPARδ in this manuscript. Please indicate if this has been shown previously.

Response: We apologize to the reviewer for our misleading words. I completely agree with you. Although some reports demonstrated that PPARδ was essential to promote fatty acid oxidation (Cheng, 2004; Palomer, 2016), there were few attestations to show free fatty acids directly activated PPARδ. In our study, FA did not increase the transcript expression of PPARδ (data not shown). Thus, we rewritten manuscript to as follow;

In the present study, we demonstrated that free fatty acids suppressed cardiomyocyte proliferation through the activation of PPARδ, …

Comment #5: If there is publicly available screening data using PPARs for cardiomyocytes, it might be interesting to see the difference between them to make the observed differences in this study.

Response: We thank the reviewer for the valuable comment. As described in Line 504, we discussed this point considering the RNA-seq data using iPS-CMs treated with a PPARδ agonist, reported by Wickramasinghe et al.

---

## [Decision Letter · Decision Letter 1]

13 Jan 2025

Fatty acid metabolism suppresses neonatal cardiomyocyte proliferation by increasing PDK4 and HMGCS2 expression through PPARδ

PONE-D-24-32166R1

Dear Dr. Tanaka,

We’re pleased to inform you that your manuscript has been judged scientifically suitable for publication and will be formally accepted for publication once it meets all outstanding technical requirements.

Kind regards,

Koh Ono, M.D., Ph.D.

Academic Editor

PLOS ONE

Additional Editor Comments (optional):

The authors addressed all concerns raised in the previous reviews. There is no further comment.

Reviewers' comments:

Reviewer's Responses to Questions

**Comments to the Author**

1. If the authors have adequately addressed your comments raised in a previous round of review and you feel that this manuscript is now acceptable for publication, you may indicate that here to bypass the “Comments to the Author” section, enter your conflict of interest statement in the “Confidential to Editor” section, and submit your "Accept" recommendation.

Reviewer #1: All comments have been addressed

Reviewer #2: All comments have been addressed

2. Is the manuscript technically sound, and do the data support the conclusions?

Reviewer #1: (No Response)

Reviewer #2: Yes

3. Has the statistical analysis been performed appropriately and rigorously? 

Reviewer #1: (No Response)

Reviewer #2: Yes

4. Have the authors made all data underlying the findings in their manuscript fully available?

Reviewer #1: (No Response)

Reviewer #2: Yes

5. Is the manuscript presented in an intelligible fashion and written in standard English?

Reviewer #1: (No Response)

Reviewer #2: Yes

6. Review Comments to the Author

Reviewer #1: The authors addressed almost all concerns raised in the previous review. There is no further comment.

Reviewer #2: (No Response)

7. PLOS authors have the option to publish the peer review history of their article (what does this mean? ). If published, this will include your full peer review and any attached files.

**Do you want your identity to be public for this peer review?** For information about this choice, including consent withdrawal, please see our Privacy Policy .

Reviewer #1: No

Reviewer #2: No

---

## [Editor Report · Acceptance letter]

PONE-D-24-32166R1

PLOS ONE

Dear Dr. Tanaka,

I'm pleased to inform you that your manuscript has been deemed suitable for publication in PLOS ONE. Congratulations! Your manuscript is now being handed over to our production team.

Kind regards,

on behalf of

Dr. Koh Ono

Academic Editor

PLOS ONE